# Emergent Symbol-like Number Variables in Artificial Neural Networks

## Abstract

There is an open question of what types of numeric representations can emerge in neural systems. To what degree do neural networks induce abstract, mutable, slot-like numeric variables, and in what situations do these representations emerge? How do these representations change over the course of learning, and how can we understand the neural implementations in ways that are unified across different models' implementations? In this work, we approach these questions by first training sequence based neural systems using Next Token Prediction (NTP) objectives on numeric tasks. We then seek to understand the neural solutions through the lens of causal abstractions or symbolic algorithms. We use a combination of causal interventions and visualization methods to find that artificial neural models do indeed develop analogs of interchangeable, mutable, latent number variables purely from the NTP objective. We then ask how variations on the tasks and model architectures affect the models' learned solutions to find that these symbol-like numeric representations do not form for every variant of the task, and transformers solve the problem in a notably different way than their recurrent counterparts. We then show how the symbol-like variables change over the course of training to find a strong correlation between the models' task performance and the alignment of their symbol-like representations. Lastly, we show that in all cases, some degree of gradience exists in these neural symbols, highlighting the difficulty of finding simple, interpretable symbolic stories of how neural networks perform numeric tasks. Taken together, our results are consistent with the view that neural networks can approximate interpretable symbolic programs of number cognition, but the particular program they approximate and the extent to which they approximate it can vary widely, depending on the network architecture, training data, extent of training, and network size.

## 1 Introduction

Both biological and artificial Neural Networks (NNs) have powerful modeling abilities. We can see an example of this in biological NNs (BNNs) from the impressive capabilities of human cognition, and we can see this in artificial NNs (ANNs) where recent advances have had such great success that ANNs have been crowned the "gold standard" in many machine learning communities (Alzubaidi et al., 2021). The inner workings of NNs, however, are still often opaque. This is, in part, due to their representations being highly distributed. Individual neurons can play multiple roles within a network (Rumelhart et al., 1986; McClelland et al., 1986; Smolensky, 1988; Olah et al., 2017; 2020; Elhage et al., 2022; Scherlis et al., 2023; Olah, 2023).

Symbolic Algorithms/programs (SAs), in contrast, defined as processes that manipulate distinct, typed entities according to explicit rules and relations, can have the benefit of consistency, transparency, and generalization when compared to their neural counterparts. A concrete example of an SA is a computer program, where the variables are abstract, mutable entities, able to represent many different values, processed by well defined functions. There are many existing theories that posit the necessity of algorithmic, symbolic, processing for higher level cognition (Do & Hasselmo, 2021; Fodor & Pylyshyn, 1988; Fodor, 1975; 1987; Newell, 1980; 1982; Pylyshyn, 1980; Marcus, 2018; Lake et al., 2017). Human designed symbolic cognitive systems, however, can lack the expressivity and performance of NNs. This is apparent in the field of natural language processing where neural architectures trained on vast amounts of data (Vaswani et al., 2017; Brown et al.,

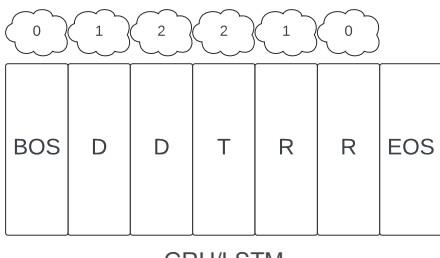 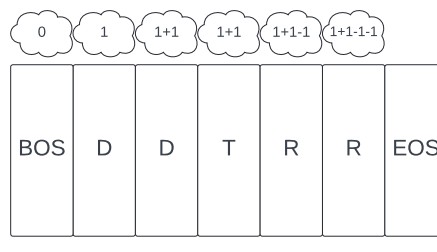

GRU/LSTM      Transformer

Figure 1: Visual depiction of different architecture's solutions achieving the same accuracy on the same numeric equivalence task. The rectangles represent token types for a task in which the model must produce the same number of R tokens followed by the EOS token as it observed D tokens before the occurence of the T token (see Methods 3.1 for more details). The thought bubbles represent causally discovered, neural variables encoded within subspaces of the models' representations. The recurrent models encode a representation of the count of the sequence that increments up before the T token and then back down after the T token to indicate the end of the task. Transformers learn a solution in which they recompute the task relevant information from their context at each step in the sequence. All NoPE transformers align with the displayed solution, where they assign opposite numeric values to the D and R tokens and then recompute their sum at each step in the sequence, knowing to stop when the difference equals 0. RoPE transformers can partially rely on positional information unless they are trained on a variant of the task that breaks number-positional correlations. In both cases, the transformers avoid using a cumulative representation of the count that is transmitted to the next step and incremented.

2020; Kaplan et al., 2020) have swept the field, surpassing the pre-existing symbolic approaches. Despite the differences between NNs and SAs, it might be argued that NNs actually implement simplified SAs; or, they may approximate them well enough that seeking neural analogies to these simplified SAs would be a powerful step toward an accessible, unified understanding of complex neural behavior. In one sense, this pursuit is trivial for ANNs, in that ANNs are implemented via computer programs. The complexity of these programs, however, can be so great that simplified SAs become be useful for understanding them. This approach of seeking to characterize neural systems in terms of *simplified* SAs is, in some sense, the goal of most cognitive science, neuroscience, and mechanistic interpretability.

In this work, we narrow our focus to numeric cognition and ask, how we can understand neural implementations of numeric concepts at the level of symbolic algorithms? Numeric reasoning has the advantage of being well studied in humans of different ages and experience levels, which provides a powerful domain for comparisons between BNNs and ANNs (Di Nuovo & Jay, 2019). We focus on a numeric equivalence task that was used to test the numeric abilities of humans whose language lacks explicit number words (Gordon, 2004). The task is formulated as a sequence of tokens, requiring the subject to produce the same number of response tokens as a quantity of demonstration tokens initially observed at the beginning of the task. This task is interesting for computational settings because the training labels vary in both type and length, and the numeric structures of interest are never explicitly labeled. Similar versions of this task have also been used in previous theoretical and computational work (El-Naggar et al., 2023; Weiss et al., 2018; Behrens et al., 2024), which provides a platform to expand upon in an effort to understand these seemingly disparate systems in unified ways.

What sorts of representations do ANNs use to solve such a task and how do they arrive at these representations? Do the networks represent numbers in a single number system? Do they use different solutions for different situations? Do the answers to these questions change over the course of training, and do the answers vary based on task and architectural details? How can we unify these solutions in satisfying ways for cognitive scientists, neuroscientists, and computer scientists alike? We wish to understand the degree to which a neural system might implement a mutable, abstract numeric variable, similar to the kind we might assign to an allocated storage location in a computer program.

In this work, we pursue these questions by first training recurrent and attention based ANNs on number related Next Token Prediction (NTP) tasks. We then perform both causal and correlative analyses to understand their neural representations and solutions. Our contributions are as follows:

1. We find causal alignments between neural variables (subspaces of the activations) and symbolic/causal variables from a counting program that increments and decrements a count variable.

2. We show that transformer architectures solve the task by referencing and recomputing information from the context at each step in the sequence, contrasted against the recurrent solution of storing a cumulative, Markovian state.

3. We show the importance of using causal interventions to substantiate claims about neural solutions, and we show the importance of finding aligned neural subspaces for the causal interventions, rather than operating directly on raw activations.

4. We show that the recurrent models' alignment to the counting program can be strongly influenced by task details that are seemingly unrelated to the underlying numeric principles.

5. We show that the symbol-like neural variables are graded, with inferior interchangeability between larger numbers and between numbers that have a greater difference in magnitude.

6. We examine the neural variables over the course of training to find a correlation between task accuracy and strength of the alignment.

7. Lastly, we show an effect of model size, where models of minimal size have a greater degree of gradience in their alignment, while larger models have more precise neural variables.

## 2 RELATED WORK

We wish to highlight the importance of using causal manipulations for interpreting neural functions in this work. Causal inference broadly refers to methods that isolate the particular effects of individual components within a larger system (Pearl, 2010). An abundance of causal interpretability variants have been used to determine what functions are being performed by the models' activations (or circuits) (Olah et al., 2018; 2020; Wang et al., 2022; Geva et al., 2023; Merrill et al., 2023; Bhaskar et al., 2024; Wu et al., 2024). Vig et al. (2020) provides an integrative review of the rationale for and utility of causal mediation in neural model analyses. We rely heavily on DAS for our analyses. This method can be thought of as a specific type of activation patching (also referred to as causal tracing) (Meng et al., 2023; Vig et al., 2020).

Many publications explore ANNs' abilities to perform counting tasks (Di Nuovo & McClelland, 2019; Fang et al., 2018; Sabathiel et al., 2020; Kondapaneni & Perona, 2020; Nasr et al., 2019; Zhang et al., 2018; Trott et al., 2018). Our tasks and modeling paradigms differ from many of these publications in that numbers are only latent in the structure of our tasks without explicit teaching of distinct symbols for distinct numeric values. El-Naggar et al. (2023) provided a theoretical treatment of Recurrent Neural Network (RNN) solutions to a parentheses closing task, and Weiss et al. (2018) explored Long Short-Term Memory RNNs (LSTMs) (Hochreiter & Schmidhuber, 1997) and Gated Recurrent Units (GRUs) (Cho et al., 2014) in a similar numeric equivalence task looking at the activations. These works showed correlates of a magnitude scaling solution in both theoretical and practically trained ANNs. Our work builds on their findings by using causal methods for our analyses, and by expanding the models considered. Lastly, we mention Behrens et al. (2024), who explored transformer counting solutions in a task similar to ours. Our work builds upon their findings by including positional encodings in our transformers, avoiding explicit labels of the numeric concepts, and providing causal analyses.

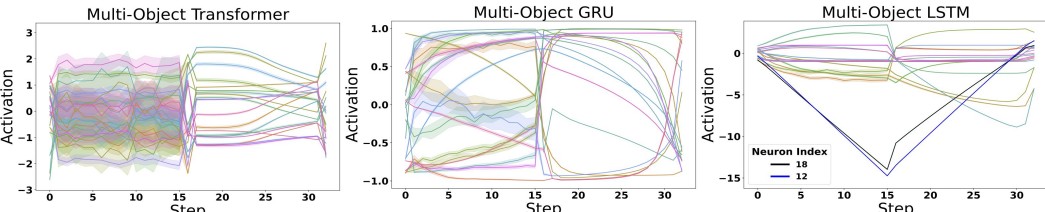

Figure 2: The activation values for each neuron (denoted by color) at each step in the trial with a object quantity of 15. Values are averaged over 15 trials. In the rightmost panel, we label the specific neurons used in a one-off causal intervention described in Sections 3.5 and 4.1.

## 3 METHODS

In this work, we train models on numeric equivalence tasks and then use interpretability methods such as Distributed Alignment Search (DAS) (Geiger et al., 2021; 2023) to understand the manner in which the models solve the task.

### 3.1 NUMERIC EQUIVALENCE TASKS

Each task we consider is defined by varying length sequences of tokens. Each sequence starts with a Beginning of Sequence (BOS) token and ends with an End of Sequence (EOS) token. Each sequence is defined by a uniformly sampled object quantity from the inclusive range of 1 to 20. The sequence is constructed as the combination of two phases. The first phase, called the demonstration phase (**demo phase**), starts with the BOS token and continues with a series of demo tokens equal in quantity to the sampled object quantity. Following the demo tokens is the Trigger token (T), indicating the end of the demo phase and the beginning of the response phase (**resp phase**). The resp phase consists of a series of resp tokens equal in number to object quantity. The EOS token follows the resp tokens, denoting the end of the sequence.

During the initial model training, we include all tokens in the autoregressive loss. During model evaluation and DAS trainings, we only consider tokens in the resp phase—which are fully determined by the demo phase. During model trainings, we hold out the object quantities 4, 9, 14, and 17. A trial is considered correct when all resp tokens and the EOS token are correctly predicted by the model after the trigger. We include three variants of this task differing only in their demo and resp token types.

**Multi-Object Task:** there are 3 demo token types {$D_1$, $D_2$, $D_3$} with a single response token type, R. The demo tokens are uniformly sampled from the 3 possible token types. An example sequence with a object quantity of 2 could be: "BOS $D_3$ $D_1$ T R R EOS"

**Single-Object Task:** there is a single demo token type, D, and a single response token type, R. An example with a object quantity of 2 is: "BOS D D T R R EOS"

**Same-Object Task:** there is a single token type, C, used by both the demo and resp phases. An example with a object quantity of 2 would be: "BOS C C T C C EOS".

For some transformer trainings, we include Variable-Length (VL) variants of each task to break count-position correlations. In these variants, each token in the demo phase has a 0.2 probability of being sampled as a unique "void" token type, V, that should be ignored when determining the object quantity of the sequence. The number of demo tokens will still be equal to the object quantity when the trigger token is presented. As an example, consider the possible sequence with a object quantity of 2: "BOS V D V V D T R R EOS".

### 3.2 MODEL ARCHITECTURES

The recurrent models in this paper consist of Gated Recurrent Units (GRUs) (Cho et al., 2014), and Long Short-Term Memory networks (LSTMs) (Hochreiter & Schmidhuber, 1997). These architectures both have a Markovian, hidden state vector that bottlenecks all predictive computations following the structure:

$$h_{t+1} = f(h_t, x_t) \tag{1}$$

$$\hat{x}_{t+1} = g(h_{t+1}) \tag{2}$$

Where $h_t$ is the hidden state vector at step $t$, $x_t$ is the input token at step $t$, $f$ is the recurrent function (either a GRU or LSTM cell), and $g$ is a multi-layer perceptron (MLP) used to make a prediction, denoted $\hat{x}_{t+1}$, of the token at step $t + 1$. We contrast the recurrent architectures against transformer architectures (Vaswani et al., 2017; Touvron et al., 2023; Su et al., 2023) in that the transformers use a history of input tokens, $X_t = [x_1, x_2, ..., x_t]$, at each time step, $t$, to make a prediction:

$$\hat{x}_{t+1} = f(X_t) \tag{3}$$

Where $f$ now represents the transformer architecture. We show results from 2 layer, single attention head transformers that use RoPE positional encodings (Su et al., 2023). Refer to Supplement A.4 and Figure 6 for more model and architectural details. We consider transformers with No Positional

Encodings (NoPE) in Supplemental section A.4. Except for in the training curves in Figure 5, we first train the models to >99.99% accuracy on their respective tasks before performing analyses. The models are evaluated on 15 sampled sequences of each of the 16 trained and 4 held out object quantities. We train 6 model seeds for each training condition. Model seeds that failed to achieve this standard were dropped from the analyses, including 3 model seeds from the LSTM models in the Same-Object task and one seed from the transformer models in each of the Single-Object and Same-Object tasks.

### 3.3 Symbolic Algorithms (SAs)

In this work, we examine the alignment of 3 different SAs to the models' distributed representations.

1. **Up-Down Program:** uses a single numeric variable, called the **Count**, to track the difference between the number of demo tokens and resp tokens at each step in the sequence. It also contains a **Phase** variable to determine whether it is in the demo or resp phase. The program ends when the Count is equal to 0 during the resp phase.

2. **Up-Up Program:** uses two numeric variables—the **Demo Count** and **Resp Count**—to track quantities at each step in the sequence. It uses a Phase variable to track which phase it is in. This program increments the Demo Count during the demo phase and increments the Resp Count during the resp phase. It ends when the Demo Count is equal to the Resp Count during the resp phase.

3. **Context Distributed (Ctx-Distr) Program:** queries a history of inputs at each step in the sequence to determine when to stop rather than encoding a cumulative quantity variable. A more specific version of this program (that appears to emerge under some conditions) is is one in which the program assigns a value of 1 to each demo token and a -1 to each resp token (or *visa-versa*) and computes their combined sum at each step in the sequence to determine the count. This program outputs the EOS token when the sum is 0.

We include Algorithms 1, 2, and 3 in the supplement which show the pseudocode used to implement the Up-Down, Up-Up, and Ctx-Distr programs in simulations. Refer to Figure 1 for an illustration of the Up-Down strategy and the more specific version of the Ctx-Distr strategy that is only observed in some transformers.

It is important to note that there are an infinite number of causally equivalent implementations of these programs. For example, the Up-Down program could immediately add and subtract 1 from the Count at every step of the task in addition to carrying out the rest of the program as previously described. We do not discriminate between programs that are causally indistinct from one another in this work.

### 3.4 Distributed Alignment Search (DAS)

DAS is a hypothesis testing framework for finding alignments between distributed systems and SAs (also referred to as causal abstractions) by performing interchange interventions (equivalently referred to as causal interventions, patches, or substitutions) (Geiger et al., 2021; 2023). For all DAS experiments, we freeze the model weights before performing the analysis.

In general, DAS measures the degree of alignment between the best subspace of a distributed model's representations with the variables from a specified SA. The method uses causal interventions to both train the alignment and to make claims about the degree of alignment. For a given variable from the SA, DAS learns an orthogonal rotation matrix, $\mathcal{R} \in R^{m \times m}$, that orients a subspace of the distributed representations along a subset of the dimensions in the representation, allowing the subspace to be freely interchanged between representations. The method relies on the notion of counterfactual behavior to train the rotation matrix. For a given SA, we know what the program's behavior should be after performing a causal intervention. This counterfactual behavior can be used as the training signal for the rotation matrices. The matrices are trained to convergence and are then validated on unseen causal interventions to determine the success of the alignment.

Concretely, we uniformly sample a time point from two separate sequences respectively. These time points are $t$ for what we will call the target sequence and $u$ for the source sequence, where *target* refers to the sequence and representations that will be intervened upon, and *source* refers to the

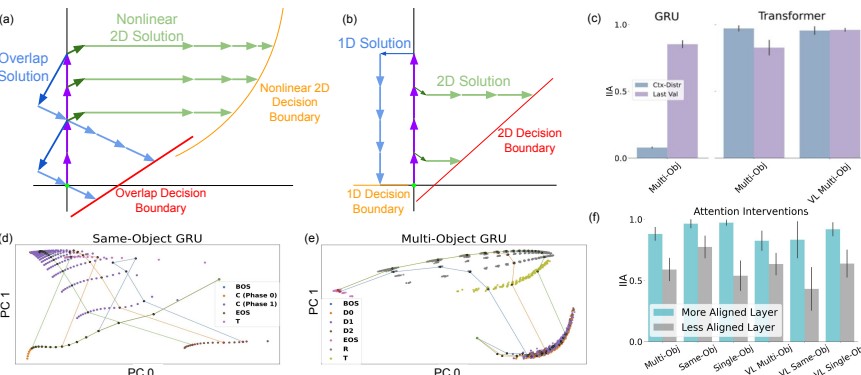

Figure 3: (a) and (b) Theoretical neural solutions to the numeric tasks. The purple arrows represent incoming demo tokens, the darker arrows indicate the trigger token, the lighter colored arrows indicate increments to the response tokens, the green dot indicates the starting point. (d) and (e) show the first two principal components of a Same-Object and Multi-Object GRUs. Multiple trajectories are shown, each point is a projected latent state in a trajectory. The lines trace individual trajectories. (See Appendix 17 and 15 for details.) (c) IIA for the full hidden state substitutions described for the Ctx-Distr program, and the DAS IIA for the Last Value alignment (see Figure 9 for expanded details). VL stands for the Variable-Length variants of the task in the x-labels. (f) IIA for the attention interventions. Results from the two layers in each model seed are sorted based on superior IIA and then averaged over seeds.

sequence and representations that will be harvested from for the intervention. We run the model on each sequence until time point $t$ and $u$ respectively. We then take the latent representations from a prespecified layer in the model at these points $t$ and $u$. We refer to these representations as the target and source vectors, $h_t^{trg} \in R^m$ and $h_u^{src} \in R^m$, where $m$ is the number of neurons in each distributed representation. We then rotate $h_t^{trg}$ and $h_u^{src}$ using $\mathcal{R}$ resulting in $r_t^{trg}$ and $r_u^{src}$, and then we replace a pre-specified number of dimensions in $r_t^{trg}$ with the same dimensions from $r_u^{src}$. Lastly we apply the inverse of the rotation to $r_u^{trg}$ resulting in a new vector, denoted $h_t^v$. This can be written formally as:

$$h_t^v = \mathcal{R}^{-1}((1-D)\mathcal{R}h_t^{trg} + D\mathcal{R}h_u^{src}) \qquad (4)$$

Where $D \in R^{m \times m}$ is a diagonal, binary matrix used to isolate the desired set of dimensions to replace. In this work, we pre-specify the number of non-zero entries in $D$ to be half of $m$. The indices of these non-zero dimensions in $D$ are unimportant as the orthogonal matrix can equivalently learn each basis in any row order. Finally, we discard $h_u^{src}$ and allow the model to continue making token predictions from point $t$ in the target sequence using $h_t^v$. We use the counterfactual behavior (tokens) of the SA as the training sequence in the autoregressive loss to train the rotation matrix.

Once our rotation matrix has converged, we can evaluate the quality of the alignment using the accuracy of the model's predictions on the counterfactual outputs in held out causal interventions. This accuracy has been referred to as the Interchange Intervention Accuracy (IIA) in previous work (Geiger et al., 2023).

For the LSTM architecture, we perform DAS on a concatenation of the $h$ and $c$ recurrent state vectors. In the GRUs, we operate on the recurrent hidden state. In the transformers, we operate on the hidden state following the first transformer layer (see Figure 6). Unless otherwise stated, we use 10000 intervention samples for training and 1000 samples for validation and testing. We uniformly sample object quantities and intervention time points, $t$ and $u$, for both the original and source sequences in the training, validation, and testing sets. We orthogonalize the rotation matrix using PyTorch's orthogonal parameterization with default settings. We train the rotation matrix for 1000, with a batch size of 512, selecting the checkpoint with the best validation performance for analysis. We use a learning rate of 0.003 and an Adam optimizer.

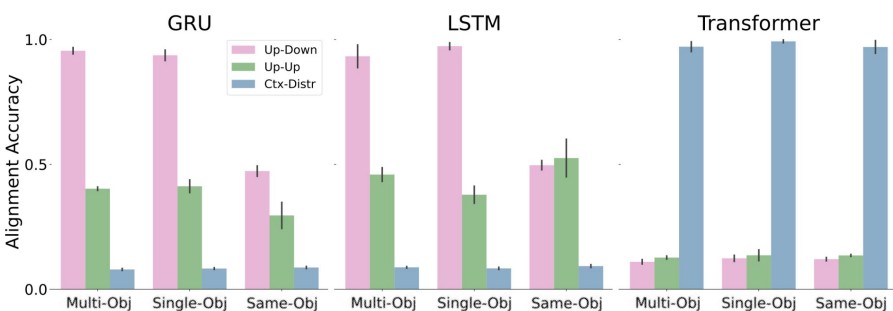

Figure 4: Interchange intervention accuracy (IIA) on variables from different symbolic programs for different tasks faceted by architecture type. The displayed IIA in the Up-Down program is taken from the Count variable. The IIA in the Up-Up program is taken as the better performing of the two possible count variables for each model type respectively. All IIA measurements show the proportion of trials in which the model successfully predicts all counterfactual R and EOS tokens following a causal intervention.

## 3.5 ADDITIONAL INTERVENTIONS

A sufficient experiment to demonstrate the lack of use of a cumulative count variable is to look for unchanged behavior after performing a full activation vector substitution on relevant hidden representations. Concretely, our main test for the Ctx-Distr strategy is to replace a full hidden state at time step $t$ with the full hidden state at time step $u$ from a different set of inputs. We provide further detail in Supplement A.5 as to why this experiment is sufficient for the claim of a time-distributed solution. We trivially apply these interventions on the recurrent hidden states in the RNNs, and we apply these interventions to the hidden states from Layer 1 in the transformer architectures. Results are displayed as Ctx-Distr in Figure 4. If the model is using the Ctx-Distr program, we would expect the models' subsequent token predictions to be unaffected by this intervention. We include a further DAS analysis to align the Last Value variable in the Ctx-Distr program (representing the increment value of the previous input token). These alignments are applied to the embeddings in the GRUs and to the embeddings that are projected into the k and v vectors in the Transformers. We leave the pre-query embeddings unperturbed, further demonstrating the anti-Markovian hidden states.

In an attempt to localize the transformers' computations to a single attention layer, we include attention interventions that directly substitute the outputs of the self-attention module from time $u$ to time $t$. We perform two intervention variants and report the average of their results in Figure 3(f). **Intervention 1:** Replace the attention output at a non-terminal step in the resp phase with the attention output taken from a terminal step. The expected counterfactual output is the EOS token. **Intervention 2:** Replace the attention output at an EOS step with the output from a non-terminal step in the resp phase. The expected prediction is a resp token.

We also explore a direct substitution of individual artificial neuron activations in the Multi-Object trained models. In these experiments, we directly substitute the activation value of a specific neuron at time step $t$ with the value of the same neuron at time step $u$ from a different sequence. We include an additional, single model activation intervention on the activations of neurons 12 and 18 from the LSTM shown in Figure 2, where we substitute both values in the interventions. In all direct interventions detailed in this section, we evaluate the model's IIA on counterfactual behavior assuming a transfer of the Count.

## 4 RESULTS

### 4.1 SYMBOLIC ALGORITHMS

Figure 4 shows DAS performance as a function of the SA used in the alignment. In the Multi-Object recurrent models, we see that the most aligned SA is the Up-Down program. The results are compared against the Up-Up program and the Ctx-Distr program which have significantly lower IIAs. We use

this as evidence in favor of the interpretation that the recurrent models develop a count up, count down strategy to track quantities within the task.

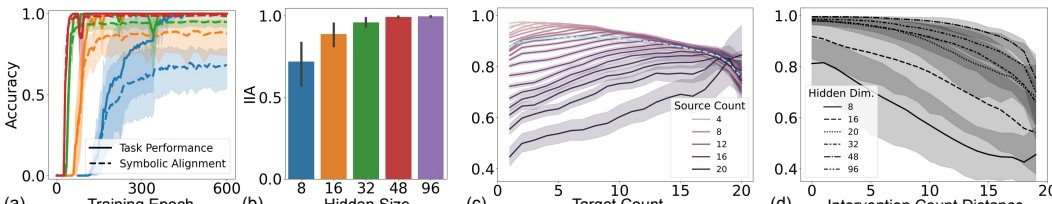

Figure 5: In all panels, the IIA comes from DAS using the Count variable in the Up-Down program on held out data. The models are all Multi-Object GRUs. (a) Both task accuracy and IIA over the course of training for different sizes of the recurrent state. (b) Converged IIA for the GRUs as a function of increasing hidden state sizes. (c) The DAS IIA where the x-axis shows the target count (the count before the intervention) and the colors denote the source count (the count that is transferred into the representation during the intervention). The curves are averaged over all models considered in panel (b). The cyan, dashed line represents the mean IIA over all interventions for a given target count—highlighting the greater number of samples for interventions on smaller numbers. (d) DAS IIA as a function of the absolute difference between the target and source counts. The line styles indicate different model sizes. Both panels (c) and (d) show that the contents of the interventions smoothly affect the IIA.

To determine how the RoPE transformers perform the task, we first look at the attention weights for both of its two layers (see Figure 10). The resp and EOS queries give surprisingly little attention to the resp tokens. We perform substitutions of non-terminal hidden states in the response phase to find that the model's predictions are largely unaffected. The results of these interventions are the Ctx-Distr bars in Figure 4. We include an additional DAS analysis on the Last Value variable from the specific version of the Ctx-Distr program in the GRUs and RoPE Transformers. The resulting IIA for these Multi-Object transformers was a value of 0.827. We also examine a set of transformers trained on the Variable-Length variant of the Multi-Object task to break count-position correlations. These Variable-Length transformers achieved an IIA of 0.960 for the same DAS analysis (see Figure 9). The lower IIA of the Multi-Object transformers is consistent with the notion that they rely, in part, on a positional readout to solve the task. In an attempt to elucidate the processing layer in which the distributed counting operation occurs, we included direct attention interventions (see Figure 3(f)). These interventions show the degree to which the EOS decision can be localized to a single attention head. The lower IIAs for the Variable-Length transformers is consistent with an interpretation that they have a stronger tendency to spread their EOS decision across both layers. We provide an additional theoretical analysis with simulations of 1 layer No Positional Encoding (NoPE) transformers in Supplement A.4 where we show that we can add and subtract from the transformer's predicted count using the strength-value of the demo tokens to add and the resp tokens to subtract.

We performed direct substitutions of individual activation values in the models' representations. Of all the neurons and models we analyzed, the best IIA was 0.399. This IIA was achieved in the LSTM model where we intervened on both the activations for neurons 12 and 18 shown in Figure 2. We use Figure 2 to highlight the difficulty of directly analyzing neural activations, and the importance of learning the rotation in DAS. Interpreting and intervening on the raw activations can be difficult and be misleading.

## 4.2 TASKS

An interesting result is the impact of demonstration token type on the resulting alignment of the recurrent models with the Up-Down program. Figure 4 shows that recurrent models trained on the Same-Object task—in which the demo tokens are the same type as the resp tokens—have poor alignment with any of the proposed SAs. We use this result to highlight the significance of the unified, interchangeable numeric representations found in the Multi-Object and Single-Object tasks.

We present a number of theoretical neural solutions to the counting task in Figure 3 as examples of possible neural solutions to each of the tasks. The Overlap Solution, shown in blue in Panel 3(a), is an

example of how some solutions may fail to align with the Up-Down solution. In the Overlap Solution, we see that the Count is entangled with the phase of the trial due to the overlap of the trajectory on the vertical axis. In this model, we would be unable to distinguish between a count of $n$ in the demo phase and a count of $n+1$ in the response phase at the overlapping points in the trajectories. We do not make claims that this is how the Same-Object models are solving the task, but merely provide the theoretical models as ways that it could solve the task.

### 4.3 MODEL SIZE, LEARNING TRAJECTORIES, AND SYMBOLIC GRADIENCE

Figure 5 shows that although many model sizes can solve the Multi-Object task, increasing the number of dimensions in the hidden states of the GRUs improves IIA in alignments with the Up-Down program. We can also see in Figure 5 that the larger models tend to have less graded alignments. We examine the symbolic alignments over the course of training in Figure 5. Of note is the correlation between alignment and performance. This is especially pronounced in the larger models. And we note the relatively flat curves of the alignment trajectories after the models solve the task.

We now provide a deeper analysis of the symbolic alignments with neural systems, where we highlight the graded nature of the neural symbols. Figure 5 shows that the GRU models trained on the Multi-Object task have worse IIA when the quantities involved in the intervention are larger, and when the intervention quantities have a greater absolute difference. We point out that the task training data forces the models to have more experience with smaller numbers, as they necessarily interact with smaller numbers every time they interact with larger numbers. This is perhaps a causal factor for the more graded representations at larger numbers. The DAS training data suffers from a similar issue, where we use a uniform sampling of the object quantities that define the training sequences and then we uniformly sample the intervention indices from these sequences. This results in a disproportionately large number of training interventions containing smaller values.

## 5 DISCUSSION/CONCLUSION

In this work we used causal methods to demonstrate the existence of symbol-like number variables within NN solutions to numeric equivalence tasks. We showed that these numeric neural variables emerge purely from an NTP objective and represent abstract information that is only latent in the task structure. These findings are a proof of principle that neural systems do not need explicit exposure to discrete numeric symbols nor built in counting principles for symbol-like representations of number to emerge.

We also demonstrated differences in the high-level solutions used by different model architectures in different tasks. Namely, we showed that increasing the dimensionality of the GRUs improved their symbolic alignment, we showed that transformers solved the tasks by recomputing relevant information at each step in the sequence—contrasted against the cumulative count variables in the recurrent models—and we showed that different solutions arise in the Same-Object Task compared to the Multi-Object and Single-Object variants. An interesting phenomenon in the LLM literature is the effect of model scale on performance (Brown et al., 2020; Kaplan et al., 2020). Although our scaling results are for GRUs on toy tasks, they are provocative for understanding why size might improve autoregressive results. Perhaps increased dimensionality allows the models to find more symbol-like, disentangled solutions when solving their NTP objectives. This is consistent with the early learning and strong correlation between performance and symbolic alignment demonstrated in larger models in Figure 5. We conjecture the possibility that this result can be explained by the lottery ticket hypothesis (Frankle & Carbin, 2019) combined with lazy learning dynamics (Jacot et al., 2020). Perhaps the majority of what these models learn are linear functions of their initial features, and increasing the dimensionality of the model increases the number of potential pathways/features that the model can use to solve the task.

We are unsure if the "stateless", time-distributed solution exhibited by the transformers generalizes beyond the counting tasks presented in this work. It is possible that this finding is representative of a more general principle—that transformers avoid solutions that use cumulative, Markovian state variables. We provide an analysis in Supplement A.4 of a one-layer transformer without positional encodings trained on a variant of the Single-Object task without a BOS token, and without a T token. We experimentally and mathematically support the idea that this model solves the task by

assigning opposite numeric values to the demo and resp tokens and averaging their values at each step in the attention. From the relatively low alignment with the Last Value variable in Figure 3(c), it seems as though the Multi-Object RoPE transformers might rely, in part, on a positional readout. We managed to get a much higher alignment when using transformers trained on a variant of the task that breaks correlations between the position and Count of the sequence. We find it worth noting that the Ctx-Distr solution exhibited by the transformers lends itself to the type of solutions that might be predicted by RASP-L (Zhou et al., 2023).

GRUs and LSTMs trained on the Same-Object Task failed to align with any of the SAs that we presented in this paper. To address this, we included Figure 3 showing the first two principal components of a Same-Object GRU model over different trial trajectories. We included theoretical models as examples of why some neural solutions might align with some SAs whereas others might not. We note that SAs that use memorization could trivially align with each of the recurrent models. One such solution might consist of a single variable that maps a tuple of the Count-Phase combination to a prediction. In this case, DAS would simply learn to transfer the complete state at each causal intervention. We are only concerned with solutions that are causally distinct from one another. We leave a more thorough, causal analysis of the Same-Object models to future work.

An important contribution of our work is in demonstrating the potential for misleading conclusions in the absence of causal analysis methods. We can see this in Figure 2 where a subset of the activations for the LSTM might be mistaken as sufficient causal features to change the model's count. Similarly, the PCA projections in Figure 3 might fail to provide predictions of neural alignment, and the attention weights shown in Figures 10- 13 might mislead on token value interchangeability. We wish to be clear, however, that these non-causal techniques are still fruitful as tools for scientific exploration and conceptualization, complementing causal methods.

We now expand upon the learning trajectories displayed in Figure 5. We can see from the performance curves that both the models' task performance and IIA begin a transition away from 0% at similar epochs and plateau at similar epochs. This result can be contrasted with an alternative result in which the alignment curves significantly lag behind the task performance of the models. Alternatively, there could have been a stronger upward slope of the IIA following the initial performance jump and plateau. In these hypothetical cases, a possible interpretation could have been that the network first develops more complex solutions or unique solutions for many different input-output pairs and subsequently unifies them over training. The pattern we observe instead is consistent with the idea that the networks are biased towards the simplest, unified strategies early in training. Perhaps our result is expected from works like Saxe et al. (2019) and Saxe et al. (2022) which show an inherent tendency for NNs trained via gradient descent to find solutions that share network pathways. This would provide a driving force towards the demo and resp phases sharing the same representation of a Count variable.

We demonstrated that the neural variables illuminated by DAS are not always perfectly symbolic, often exhibiting a smooth, graded influence from the content of the variables being intervened upon. We interpret these results as a reminder that representations in distributed systems exist on a continuum despite seemingly discrete, symbolic performance on tasks. These results have an analogy to children's number cognition in which children may appear to possess a symbol-like understanding of exact numbers and their associated principles, but when probed deeper, the symbol-like picture falls apart (Wynn, 1992; Davidson et al., 2012). Perhaps the graded nature of the neural variables reinforces the utility of thinking about network solutions as trajectories in a dynamical system. We use our findings as a reminder that although NNs may discover approximations to interpretable, symbol-like solutions, their representations are still ultimately graded—adding nuance to the effort of SA alignment.

We conclude by noting that it is, by definition, always possible to represent an ANN with a SA due to the fact that ANNs are implemented using computer (symbolic) programs. Our goal of NN-SA alignment is to find simplified, unified ways of understanding complex ANNs. If an ANN has poor alignment for a specific region of the symbolic variables, we argue that the SA simply needs to be refined. In our case, any lack of alignment for numbers beyond the training range of 20 can be solved by adding a limit to the Count variable in the Up-Down program. Any choice of SA refinement is dependent on the goals of the work. We leave further refinements to the algorithms presented in this work to future directions.

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

# A APPENDIX / SUPPLEMENTAL MATERIAL

## A.1 ADDITIONAL FIGURES

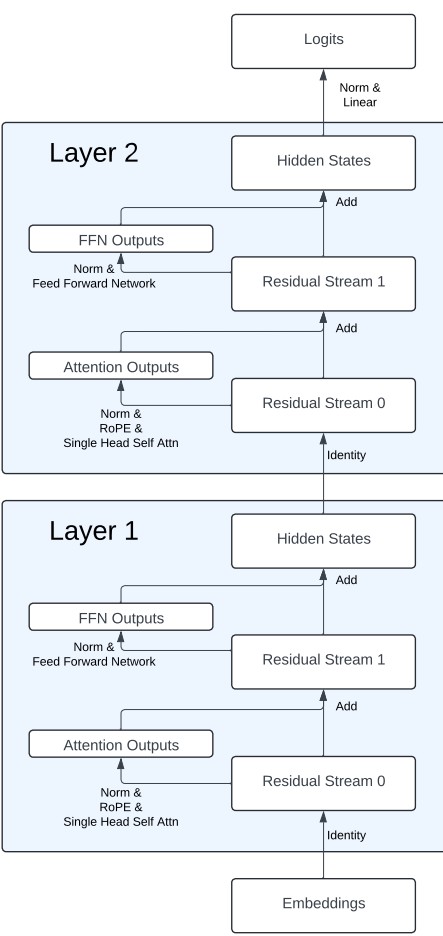

Figure 6: Diagram of the main transformer architecture used in this work. The white rectangles represent activation vectors. The arrows represent model operations. Unless otherwise stated, all interchange interventions were performed on the Hidden State activations from Layer 1 or the Residual Stream 0 within Layer 1 for the key and value projections. All normalizations are Layer Norms (Ba et al., 2016).

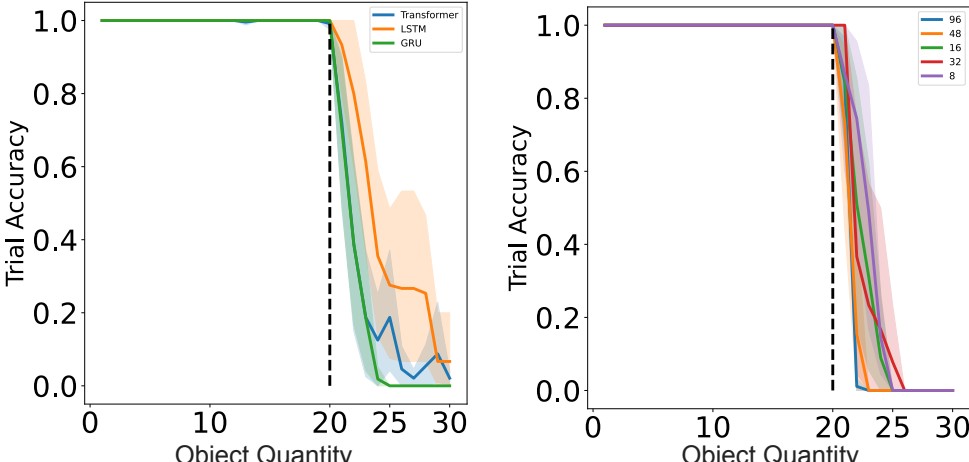

Figure 7: *Left:* The model performance on the tasks. This result includes the Multi-Object, Single-Object, and Same-Object tasks. Each object quantity includes 15 sampled sequences (even when only one configuration exists for that object quantity). 3 model seeds were dropped from the LSTM models in the Same-Object task due to lower than 99% accuracy. One seed was dropped from the transformer models in each the Single-Object and Same-Object tasks for the same reason. *Right:* The GRU performance on the tasks facetted by model size (hidden dimensionality). This result is only for GRUs train on the Multi-Object task.

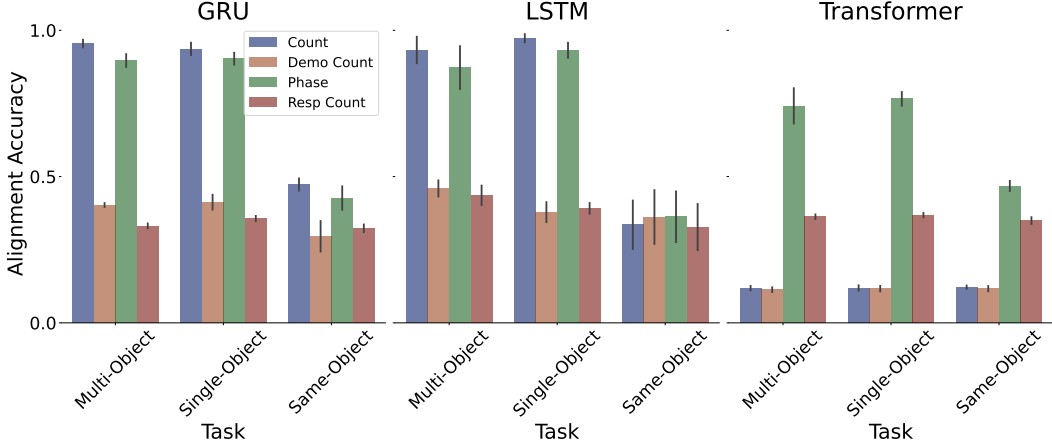

Figure 8: Interchange intervention accuracy (IIA) on variables from different symbolic programs for different tasks faceted by architecture type. The y-axis shows the proportion of trials in which the model predicts all counterfactual tokens correctly after a causal intervention for the corresponding variable on held out data.

## A.2 MODEL DETAILS

All artificial neural network models were implemented and trained using PyTorch (Paszke et al., 2019) on Nvidia Titan X GPUs. Unless otherwise stated, all models used an embedding and hidden state size of 20 dimensions. To make the token predictions, each model used a two layer multi-layer perceptron (MLP) with GELU nonlinearities, with a hidden layer size of 4 times the hidden state dimensionality with 50% dropout on the hidden layer. The GRU and LSTM model variants each consisted of a single recurrent cell followed by the output MLP. Unless otherwise stated, the transformer architecture consisted of two layers using Rotary positional encodings (Su et al., 2023). Each model variant used the same learning rate scheduler, which consisted of the original transformer (Vaswani et al., 2017) scheduling of warmup followed by decay. We used 100 warmup steps, a maximum learning rate of

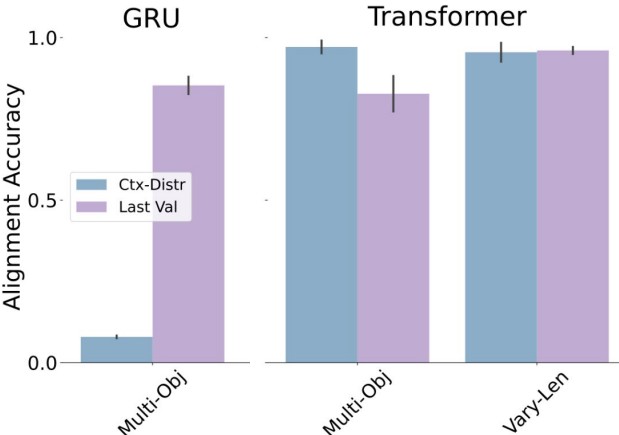

Figure 9: Interchange intervention accuracy (IIA) comparing the Ctx-Distr results from the GRU and Transformer architectures displayed in Figure 4 with the DAS alignment to the Last Value variable. We include results from a transformer trained on the Variable-Length version of the Multi-Object Task. The Ctx-Distr interventions consist of full replacements of the hidden states to determine the degree to which the models accumulate a state encoding of the important information for the task. The Last Value variable is a value of +1, -1, or 0 assigned to each incoming token. We apply DAS on the model embeddings, and only to the embeddings leading into the key and value projections in the transformers. We can see that although the Variable-Length and Multi-Object transformers both use an anti-Markovian solution (they avoid using a cumulative state) as demonstrated by the Ctx-Distr interventions, the Variable-Length transformers align much better to the Last Value variable. This is consistent with an interpretation in which the Multi-Object transformers rely, to some degree, on a positional encoding readout. This reliance is broken when the task breaks the correlation between position and count. We include the GRU results to show that the GRUs also, to some degree, assign a numeric value to each incoming embedding independent of the phase.

0.001 , a minimum of 1e-7, and a decay rate of 0.5. We used a batch size of 128, which caused each epoch to consist of 8 gradient update steps.

## A.3 DAS Training Details

### A.3.1 Rotation Matrix Training

To train the DAS rotation matrices, we applied PyTorch's default orthogonal parametrization to a square matrix of the same size as the model's state dimensionality. PyTorch creates the orthogonal matrix as the exponential of a skew symmetric matrix. In all experiments, we selected the number of dimensions to intervene upon as half of the dimensionality of the state. We chose this value after an initial hyperparameter search that showed the number of dimensions had little impact on performance between 5-15 dimensions. We sampled 10000 sequence pairs and for each of these pairs, we uniformly sampled corresponding indices to perform the interventions. We excluded the BOS, and EOS tokens from possible intervention sample indices. When intervening upon a state in the demo phase, we uniformly sampled 0-3 steps to continue the demo phase before changing the phase by inserting the trigger token. We used a learning rate of 0.003 and a batch size of 512.

### A.3.2 Symbolic Program Algorithms

## A.4 Simplified Transformer

The self-attention calculation for a single query $q_r \in R^d$ from a response token, denoted by the subscript $r$, is as follows:

$$\text{Attention}(q_r, K, V) = V\left(\text{softmax}\left(\frac{K^\top q_r}{\sqrt{d}}\right)\right) = \sum_{i=1}^{n} \frac{e^{\frac{q_r^\top k_i}{\sqrt{d}}}}{\sum_{j=1}^{n} e^{\frac{q_r^\top k_j}{\sqrt{d}}}} v_i = \sum_{i=1}^{n} \frac{s_i^r}{\sum_{j=1}^{n} s_j^r} v_i \quad (5)$$

---

**Algorithm 1** One sequence step of the Up-Down Program

$q \leftarrow$ Count
$p \leftarrow$ Phase
$y \leftarrow$ input token
**if** $y ==$ BOS **then**                                                    ▷ BOS is beginning of sequence token
    $q \leftarrow 0, p \leftarrow 0$
    return sample(D)                                         ▷ sample a demo token
**else if** $y \in$ D **then**                                              ▷ D is set of demo tokens
    $q \leftarrow q + 1$
    return sample(D)
**else if** $y ==$ T **then**                                               ▷ T is trigger token
    $p \leftarrow 1$
**else if** $y ==$ R **then**                                               ▷ R is response token
    $q \leftarrow q - 1$
**end if**
**if** $(q == 0) \, \& \, (p == 1)$ **then**
    return EOS                                               ▷ EOS is end of sequence token
**end if**
return R

---

**Algorithm 2** One sequence step of the Up-Up Program

$d \leftarrow$ Demo Count
$r \leftarrow$ Resp Count
$p \leftarrow$ Phase
$y \leftarrow$ input token
**if** $y ==$ BOS **then**                                                  ▷ BOS is beginning of sequence token
    $d \leftarrow 0, r \leftarrow 0, p \leftarrow 0$
    return sample(D)                                         ▷ sample a demo token
**else if** $y \in$ D **then**                                              ▷ D is set of demo tokens
    $d \leftarrow d + 1$
    return sample(D)
**else if** $y ==$ T **then**                                               ▷ T is trigger token
    $p \leftarrow 1$
**else if** $y ==$ R **then**                                               ▷ R is response token
    $r \leftarrow r + 1$
**end if**
**if** $(d == r) \, \& \, (p == 1)$ **then**
    return EOS                                               ▷ EOS is end of sequence token
**end if**
return R

---

**Algorithm 3** One sequence step of the specific Ctx-Distr Program

---

$v \leftarrow$ list of previous values excluding the most recent step
$\ell \leftarrow$ Last Value $\qquad \triangleright$ The value of the most recent token
$p \leftarrow$ Phase $\qquad \triangleright$ 0 indicates the demo phase, 1 is the response phase
$y \leftarrow$ input token

$v$.append($\ell$)
$s \leftarrow$ SUM($v$)
**if** $y ==$ BOS **then** $\qquad \triangleright$ BOS is beginning of sequence token
$\quad \ell \leftarrow 0, p \leftarrow 0$
$\quad$ **return** sample(D) $\qquad \triangleright$ sample a demo token
**else if** $s \leq 0$ and $p == 1$ **then** $\qquad \triangleright$ Sum is 0 or less in the response phase
$\quad$ **return** EOS $\qquad \triangleright$ EOS is end of sequence token
**else if** $y ==$ T or $y ==$ R **then** $\qquad \triangleright$ T is trigger token, R is response token
$\quad p \leftarrow 1$
$\quad \ell \leftarrow -1$
$\quad$ **return** R
**else if** $y \in$ D **then** $\qquad \triangleright$ D is set of demo tokens
$\quad \ell \leftarrow 1$
**end if**

**if** $p == 1$ **then**
$\quad$ **return** R
**else**
$\quad$ **return** sample(D)
**end if**

---

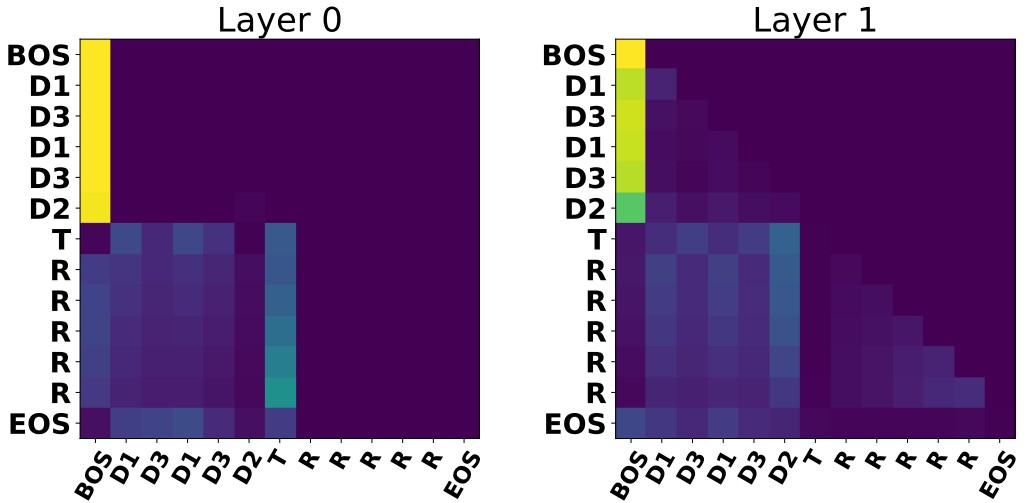

Figure 10: Attention weights for a single transformer with two layers using rotary positional encodings trained on the Multi-Object Task. Queries are displayed on the vertical axis in order of their appearance starting at the top. Keys are displayed on the horizontal axis starting from the left. Queries are only able to attend to themselves and preceding keys.

Where $d$ is the dimensionality of the model, $n$ is the sequence length, $K \in R^{d \times n}$ is a matrix of column vector keys, $V \in R^{d \times n}$ is a matrix of column vector values, and $s_i^r = e^{\frac{q_r^\top k_i}{\sqrt{d}}}$, using $r$ to denote the token type that produced $q$. We refer to $s_i^r v_i$ as the strength value of the $i^{\text{th}}$ token for the query $q_r$.

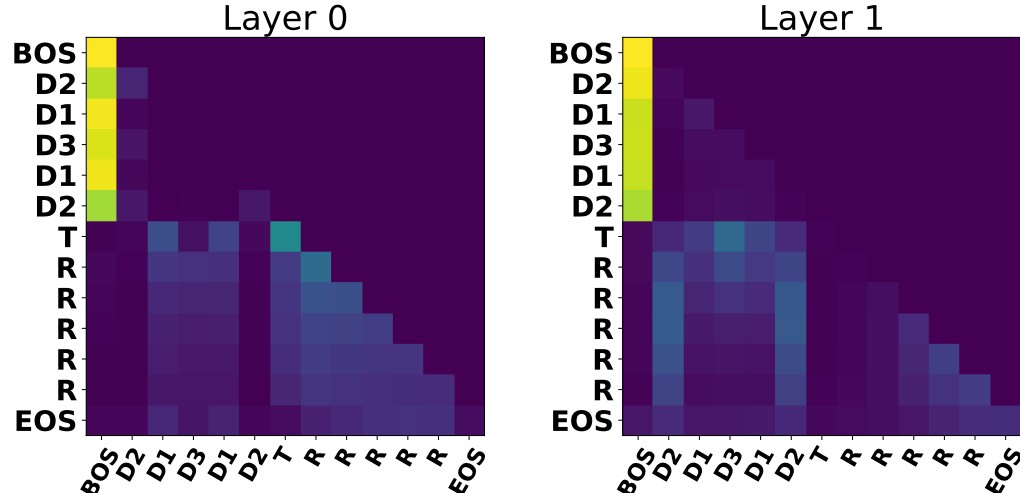

Figure 11: Attention weights for a single transformer with two layers using rotary positional encodings trained on the Variable-Length variant of the Multi-Object Task. Queries are displayed on the vertical axis in order of their appearance starting at the top. Keys are displayed on the horizontal axis starting from the left. Queries are only able to attend to themselves and preceding keys.

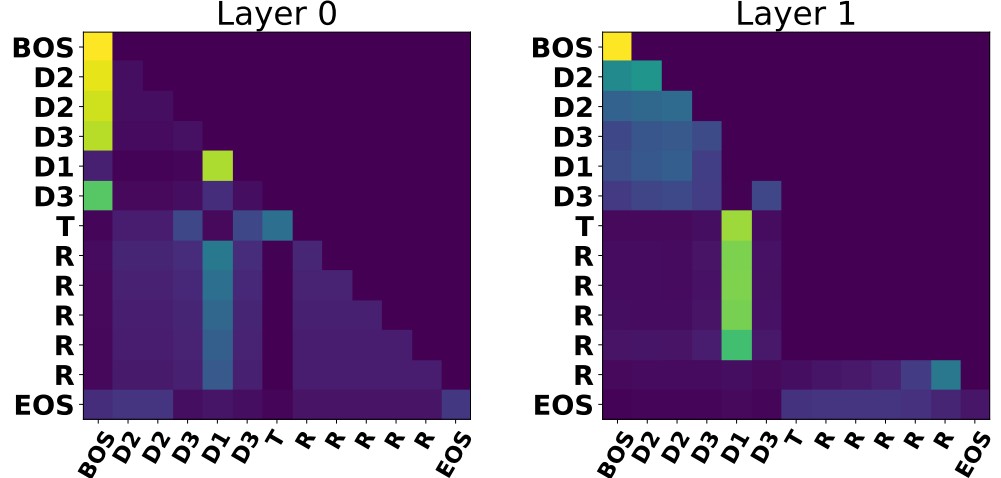

Figure 12: Attention weights for a single transformer model seed with two layers and no positional encodings (NPE) trained on the Multi-Object Task. Queries are displayed on the vertical axis in order of their appearance starting at the top. Keys are displayed on the horizontal axis starting from the left. Queries are only able to attend to themselves and preceding keys.

In a transformer without positional encodings, each of the queries for the response tokens will produce equal strength values to one another for a given key-value pair. Thus, under the assumption that the attention mechanism is performing a sum of the count contributions from each token in the sequence, we should be able to use the $s_i^r v_i$ to increment and decrement the number of tokens the model will produce for a given sequence in the following way:

$$\text{IncrementedAttention}(q_r, K, V) = \frac{1}{s_r^r + \sum_{j=1}^{n} s_j^r} \left( s_r^r v_r + \sum_{i=1}^{n} s_i v_i \right) \qquad (6)$$

Where the subscript $r$ denotes the strength $s_r$ and value $v_r$ were calculated from a response key-value pair. Similarly, we can decrement the count using a key-value pair from a demonstration token, D, in

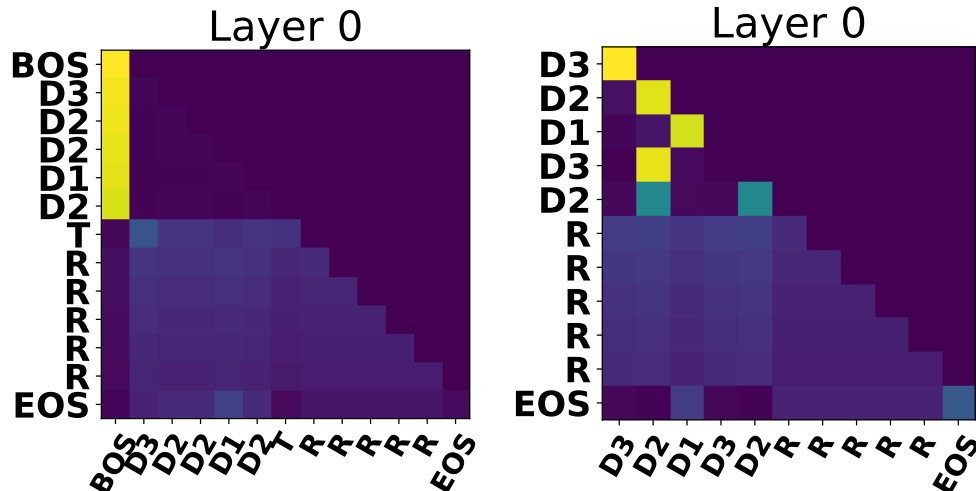

Figure 13: Left: Attention weights for a single transformer model seed with one layer and no positional encodings. Right: Attention weights for a single transformer seed with one layer and no positional encodings trained without the BOS and trigger token types. In both figures, queries are displayed on the vertical axis in order of their appearance in the sequence starting at the top. Keys are displayed on the horizontal axis starting from the left. Queries are only able to attend to themselves and preceding keys.

the following way.

$$\text{DecrementedAttention}(q_r, K, V) = \frac{1}{s_D^r + \sum_{j=1}^n s_j^r}\left(s_D^r v_D + \sum_{i=1}^n s_i v_i\right) \tag{7}$$

As a sanity check we use single layer transformers without positional encodings and add and subtract from the transformer's count using the strength values as described in this section. We are able to change the position at which it produces the EOS token with 100% accuracy.

## A.5 ADDITIONAL INTERVENTIONS CONTINUED

We detail in this section why our activation transfers are sufficient to demonstrate that the transformers use a solution that re-references/recomputes the relevant information to solve the tasks at each step in the sequence. The hidden states in Layer 1 are a bottleneck at which a cumulative counting variable must exist if it were to use a strategy like the Up-Down or Up-Up programs. This is because the Attention Outputs of Layer 1 are the first activations that have had an opportunity to cross communicate between token positions. This means that the representations between the Residual Stream 1 of Layer 1 up to the Residual Stream 0 of Layer 2 cannot have read off a cumulative state from the previous token position other than reading off the positional information from the previous positional encodings. The 2-layer architecture is then limited in that it has only one more opportunity to transfer information between positions—the attention mechanism in Layer 2. Thus, if a hidden state at time $t$ were to have encoded a cumulative representation of the count that will be used by the model at time $t + 1$, that cumulative representation must exist in the activation vectors between the Residual Stream 1 in Layer 1 and the Residual Stream 0 of Layer 2. If it is using such a cumulative representation, then when we perform a full activation swap in the Layer 1 hidden states then the resulting predictions should be influenced by the swap. As Figures 4 and 14 indicate, the resulting transformer predictions are mostly unchanged by the intervention, demonstrating a recomputing of information at each step in the task.

## A.6 VARIABLE-LENGTH TASK VARIANTS

Here we include additional tasks to prevent the transformers with positional encodings from learning a solution that relies on reading out positional information. We introduce Variable-Length variants of

each of the Multi-Object, Single-Object, and Same-Object tasks. In the Variable-Length versions, each token in the demo phase has a 0.2 probability of being sampled as a unique "void" token type, V, that should be ignored when determining the object quantity of the sequence. The number of demo tokens will still be equal to the object quantity when the trigger token is presented. We include these void tokens as a way to vary the length of the demo phase for a given object quantity, thus breaking correlations between positional information and object quantities. As an example, consider the possible sequence with a object quantity of 2: "BOS V D V V D T R R EOS".

We show the transformer performance and the IIA for the Ctx-Distr interventions in Figure 14. Although we do not make strong claims about the manner in which these transformers solve these new tasks, we do highlight the fact that the transformers can no longer use a direct positional encoding readout to achieve 100% accuracy. These results are consistent with the hypothesis that the transformers are using the more specific, summing version of the Ctx-Distr strategy to solve these tasks, much as the no-positional encoding transformers do.

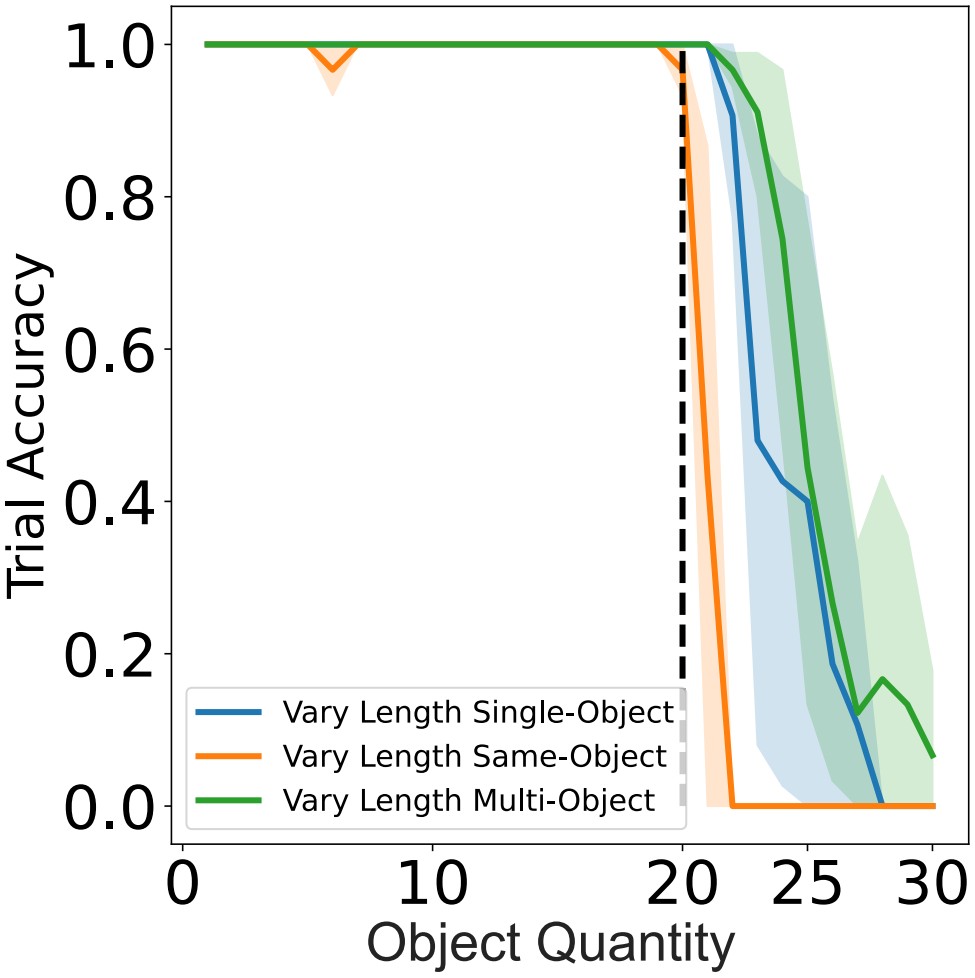

Figure 14: *Left:* The transformer performance on variable length variants of the 3 tasks. *Right:* The interchange intervention accuracy using the Ctx-Distr program for the transformer models on the variable length tasks. In both panels, 4 model seeds were dropped from the models in the variable length Same-Object task due to lower than 99% accuracy, and one seed was dropped from the variable length Single-Object task for the same reason.

## A.7 PRINCIPLE COMPONENTS ANALYSIS

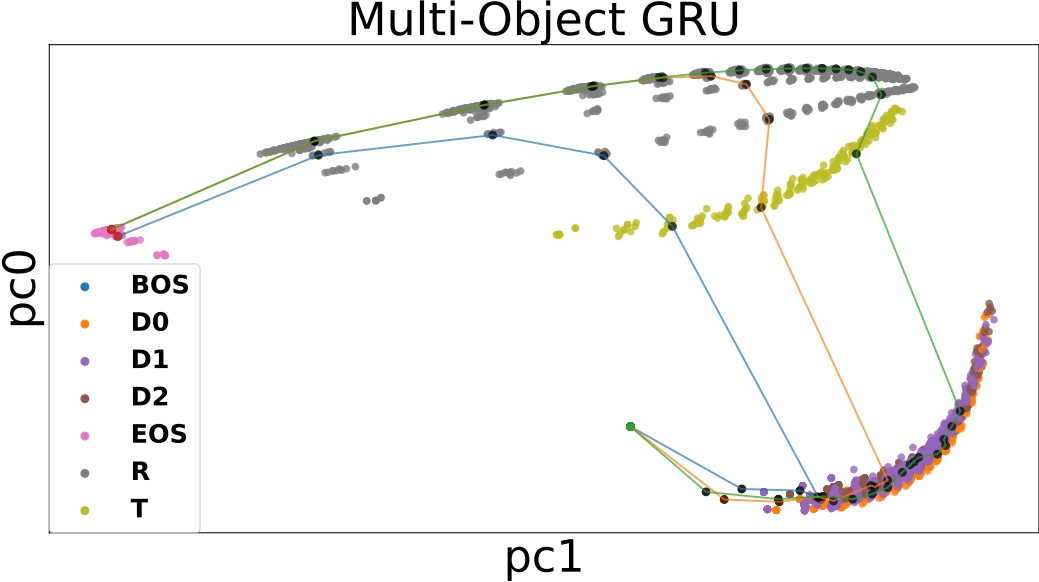

Figure 15: Principal Components Analysis of a single GRU model seed including hidden state representations over 10 trials for each object quantity from 1 to 20 in the Multi-Object task variant. Green points indicate the start of a plotted trajectory, black points indicate an intermediate step, and red points indicate the end of a plotted trajectory. The blue line plots a single trajectory from start to finish with a object quantity of 3. Similarly, the orange and green lines follow single trajectories of 7 and 15 respectively.

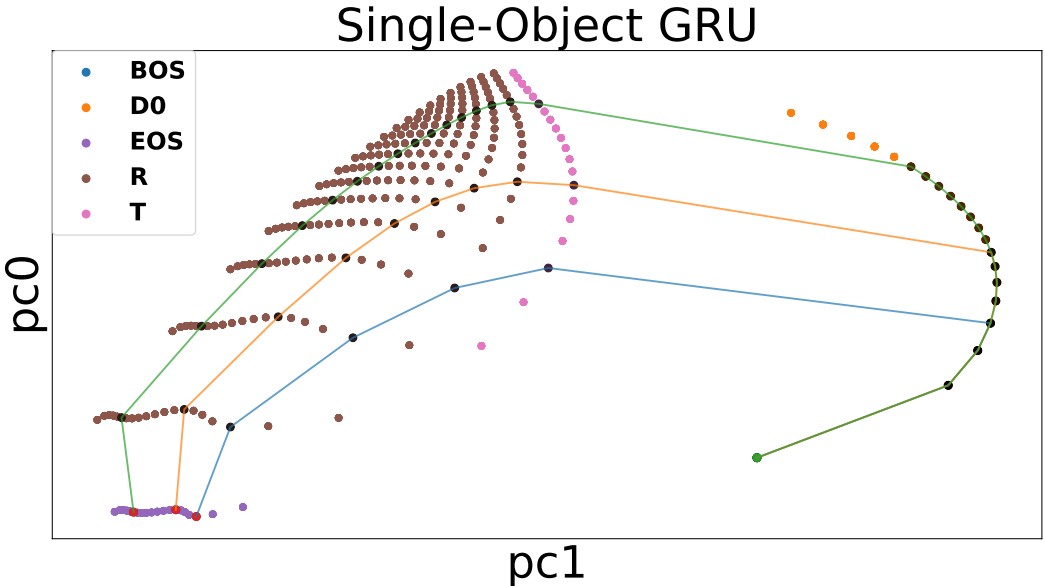

Figure 16: Principal Components Analysis of a single GRU model seed including hidden state representations over 10 trials for each object quantity from 1 to 20 in the Single Object task variant. Green points indicate the start of a plotted trajectory, black points indicate an intermediate step, and red points indicate the end of a plotted trajectory. The blue line plots a single trajectory from start to finish with a object quantity of 3. Similarly, the orange and green lines follow single trajectories of 7 and 15 respectively.

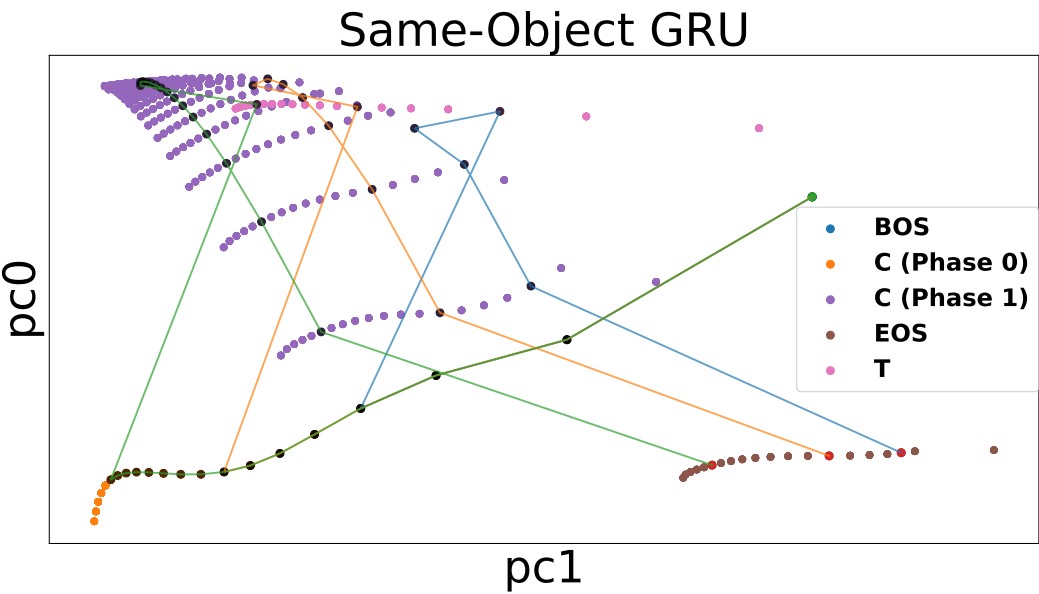

Figure 17: Principal Components Analysis of a single GRU model seed including hidden state representations over 10 trials for each object quantity from 1 to 20 in the Same-Object task variant. Green points indicate the start of a plotted trajectory, black points indicate an intermediate step, and red points indicate the end of a plotted trajectory. The blue line plots a single trajectory from start to finish with a object quantity of 3. Similarly, the orange and green lines follow single trajectories of 7 and 15 respectively.

