# OpenReview forum: "Emergent Symbol-Like Number Variables in Artificial Neural Networks"
_ICLR.cc/2025/Conference — Submitted to ICLR 2025_

### Official Review · Reviewer_J2Vq · 2024-10-18

**Soundness:** 3
**Presentation:** 3
**Contribution:** 2
**Rating:** 5
**Confidence:** 3

**Summary:**

The paper investigates the emergence of symbol-like number variables in the representations of sequence models trained on next-token prediction. The task is such that latent number variables are presumably required to solve it in the general case, but such variables are never given as explicit supervision targets and must be learned purely from the next-token prediction task pressure. This means that the findings have particular relevance to, for instance, language models. The methods used to investigate the presence or absence of number variables are taken from causality and mechanistic interpretability. The main finding is that simple RNNs do indeed appear to learn the correct symbol-like number variables on most tasks, albeit in an imperfect way.

My current leaning is to reject the paper, but I am willing to increase my score to a marginal accept if weaknesses 1-5 (along with my questions) are all addressed in the paper or satisfactorily rebutted.

**Strengths:**

1. The work is, for the most part, easy to follow. I understood the tasks, methods (although seem points below on DAS), and results.
2. The core claims appear to be well-supported.
3. I like the approach of using methods from causality and training using next-token prediction as opposed to more explicit pressure for symbolic number representations. The advantages of these methods are clearly articulated.

**Weaknesses:**

1. As far as I can tell, the DAS interventions only really test within-distribution interventions. Despite some target quantities being held out, these are basically holes within an observed range of integers. Why not see if the model generalizes to numbers outside this range (e.g., 30) and at what value things begin to break down? Also, why not apply DAS for both the count variable and the phase variable in the Up-Down program, and then test to see if the model can be causally manipulated in more interesting ways (e.g., flipping the phase variable several times within the same sequence, even though at training time it only ever flips once and in one direction). These sorts of more aggressive causal interventions are important to understand the degree to which the learned function is symbol-like, since a hallmark of symbolic programs is their systematicity.
2. The explanation of DAS is, in my opinion, not written clearly. I understand that this was not the contribution of the current work, but given its centrality it should be explained with more intuition (including with a figure). If I had not already been familiar with DAS prior to reading this paper, I think a lot of it would have gone over my head.
3. Figure 3 right panel seems not to provide much interesting information, and I would suggest removing this result. It is obvious that increasing model size will increase task performance, and also obvious that all accuracy metrics (including symbolic alignment) should more or less improve over the course of training. Figure 3 left panel seems completely sufficient on its own to show the results of Section 4.2.
4. That the Same-Object task should yield unaligned models seems completely mysterious to me. If anything, I would have expected that models trained on Same-Object would be most aligned among the 3 tasks. Given that this paper is about mechanistic interpretability with respect to symbolic algorithms, I think it is extremely important that the strange behaviour for the Same-Object task, in which no symbolic algorithm is aligned, be explained.
5. Some ungrammatical sentences and typos peppered throughout the text (e.g., line 169, line 190, line 246, etc.) need to be corrected. Please proofread again for clarity.
6. The work lacks a bit of ambition, in my opinion. If one is going to investigate alignment to symbolic programs (which has already been done before in other settings, such as supervised learning), then why not go for more complex tasks which have more complex symbolic algorithmic solutions, with more variables that one could check alignment with respect to? I understand that this was intended as a proof of principle, but honestly the proof of principle in very simple tasks like these seems at once reminiscent of other work I have seen at various conferences, and does not tell me much about whether these sorts of results scale in practice to more complex reasoning tasks. In sum, the results seem to be a bit of a low-hanging fruit, and I would have liked to see a more ambitious project investigating alignment to more complex symbolic programs.

**Questions:**

1. On line 140, what is the “Phase variable to track quantity at each step in the sequence”? What does the word “quantity” refer to here? I’m assuming this variable tracks whether the T token has been encountered yet (as for the Up-Down Program), but the sentence is confusing and doesn’t seem to indicate that.
2. Why is DAS in the Transformer case only applied to the hidden states between the two Transformer layers, as opposed to within things like MLP hidden states?
3. Why only investigate the presence of a single program variable? For instance, in the Up-Down Program, why not try to find and causally manipulate the representation of both the count variable and the phase variable (wee “weakness” above)?
4. Regarding Section 4.4, the proposed explanation for why representations of large numbers are less precise implies that task performance should also be worse at these larger numbers. Is there evidence of this? Despite the fact that all models achieve 99.99% accuracy, perhaps you can still check if it is true at earlier points in training.

---

> ### Author Response · Authors · 2024-11-24
> **Addressing multiple concerns from the reviewer**
>
> In reference to the reviewer's Weaknesses section:
>
> 1. Yes, we enjoy and share your curiosity about questions of how well DAS can generalize outside of their training distribution, and we appreciate questions about what DAS can tell us about OOD generalization of the neural network of interest. In our case, the networks did not tend to have high OOD extrapolated accuracies on the tasks (see Figure 7) and, in preliminary experiments, there did not appear to be a correlation between DAS interchange accuracy and OOD network performance. We have thus pushed those directions to future work. And yes, we agree that more aggressive interchange interventions would make the work more powerful. We believe the claims of our paper, however, stand in the absence of more complex interventions.
>
> 2. We hope our revised section 3.4 addresses these concerns.
>
> 3. We expand on why we include the symbolic alignment over the course of training at line 515 in the Discussion section.
>
> "We can see from the performance
> curves that both the models’ task performance and alignment performance begin a transition away
> from 0% at similar epochs and plateau at similar epochs. This result can be contrasted with an
> alternative result in which the alignment curves significantly lag behind the task performance of
> the models. Alternatively, there could have been a stronger upward slope of the IIA following the
> initial performance jump and plateau. In these hypothetical cases, a possible interpretation could have
> been that the network first develops more complex solutions or unique solutions for many different
> input-output pairs and subsequently unifies them over training. The pattern we observe instead is
> consistent with the idea that the networks are biased towards the simplest, unified strategies from the
> beginning of training. Perhaps our result is to be expected in light of works like Saxe et al. (2019)
> and Saxe et al. (2022) which show an inherent tendency for NNs trained via gradient descent to find
> solutions that share network pathways. This would explain the driving force towards the demo and
> response phases sharing the same representation of a Count variable."
>
> This analysis is interesting for NN learning dynamics as it implies that the networks initially find a shared solution rather than first finding multiple disparate solutions and then combining them into one solution.
>
> 4. We agree that our initial treatment of the Same-Object models left much to be desired. We have added a number of theoretical neural models to our revised version in Figure 3 to provide examples of how the variables can be entangled, preventing alignment with the Up-Down program. We also added a PCA panel to Figure 3 to give more insight into the trained model's representations. We still fail to satisfyingly show, however, how the Same-Object models solve the task.
>
> 5. Thank you.
>
> 6. Yes, we appreciate the critique. We agree that the counting tasks and symbolic algorithms presented in our work are simple. We push back by pointing out that many of our analyses demonstrate interesting phenomena without the need for increased complexity. Complexity is useful insofar as it provides additional insights into the issues being discussed. Our claims about the non-markovian transformer solutions, the emergence and gradience of neural symbols, and the importance of causal methods are all substantiated without the need for added complexity. Again, however, we agree that added complexity would make for more satisfying work.
>
>
>
> In response to the Questions section:
> 1. Thank you, we meant that it is the same Phase variable as described in the Up-Down program. We have reworded in the most recent draft.
>
> 2. We show that the transformers use non-markovian states with the Ctx-Distr interventions. As such, DAS applied at any processing layer in the transformers using the Up-Down or Up-Up programs will have low IIA. See supplemental section A.5 for more details on this idea.
>
> 4. It is possible to have imprecise representations that are still precise enough to produce 100% accuracy on a task. We can think of the precision of a representation as referring to the margin between a point cloud representing the numeric concept and the decision boundary. If the edges of the point cloud are relatively close to a decision boundary, but never erroneously cross the boundary, they might have low precision while still being sufficient for 100% accuracy.

---

> > ### Comment · Reviewer_J2Vq · 2024-11-25
> >
> > Thank you for engaging with my feedback. Some of it has been addressed and I do like certain aspects of this work, so I have raised my rating to a 5. Here are the reasons I do not wish to raise it further:
> > 1. As I said earlier, I think the work lacks a bit of ambition. As reviewer sMWR pointed out, there is a lot of similar work on investigating the presence of arithmetic symbol-like solutions in recurrent models, and I think it's time to go beyond this an try to test alignment to more complex programs. The authors' rebuttal says more complexity is not always good for better scientific understanding, and I agree, but that is only the case if the lower-complexity proofs of concept have not already been sufficiently explored.
> > 2. My question about OOD interventions using DAS (and combinations of interventions on multiple variables at once) was aimed at understanding to what degree the model representations truly represent symbol-like number variables. If the representation cannot be intervened on OOD with correct counterfactual predictions, then the representation doesn't truly represent number. Again I want to emphasize that symbols and symbolic models are so special because they provide systematicity, so confirming that OOD interventions work is crucial to any claim about alignment between neural networks and symbolic programs.

---

> ### Author Response · Authors · 2024-11-25
>
> Thank you for being an engaging reviewer. You have helped make the paper much better.
>
> 2. This is an extremely helpful comment. We failed to convey a key philosophical contribution in our previous drafts. Our goal with the symbolic alignments is to provide a unified, simplified way of understanding the numeric representations in the neural activity. We agree that the ANNs aren't "truly" representing number symbols. A goal of our analysis on the gradience of the neural variables in Figure 5 is to prove your point. Rather, the alignment with the symbolic algorithms provides a simplified way to understand the neural system. So, when it doesn't align, the symbolic algorithm just needs to be refined. We have added the following excerpt to the introduction at line 080 to illustrate this point (where SA stands for symbolic algorithm):
>
> "Despite the differences between NNs and SAs, it might be argued that NNs actually implement
> simplified SAs; or, they may approximate them well enough that seeking neural analogies to these
> simplified SAs would be a powerful step toward an accessible, unified understanding of complex
> neural behavior. In one sense, this pursuit is trivial for ANNs, in that ANNs are implemented via
> computer programs. The complexity of these programs, however, can be so great that simplified SAs
> become be useful for understanding them."
>
> We have also added the following paragraph at the end of the discussion at line 532:
>
> "We conclude by noting that it is, by definition, always possible to represent an ANN with a SA due
> to the fact that ANNs are implemented using computer (symbolic) programs. Our goal of NN-SA
> alignment is to find simplified, unified ways of understanding complex ANNs. If an ANN has poor
> alignment for a specific region of the symbolic variables, we argue that the SA simply needs to be
> refined. In our case, any lack of alignment for numbers beyond the training range of 20 can be solved
> by adding a limit to the Count variable in the Up-Down program. Any choice of SA refinement is
> dependent on the goals of the work. We leave further refinements to the algorithms presented in this
> work to future directions."

---

### Official Review · Reviewer_bPA8 · 2024-10-27

**Soundness:** 3
**Presentation:** 4
**Contribution:** 3
**Rating:** 6
**Confidence:** 2

**Summary:**

This paper investigates how neural networks, specifically recurrent networks and Transformer architectures, can generate symbol-like numeric representations purely from the next-token prediction (NTP) objective. The study reveals that recurrent networks tend to align closely with symbolic counting processes, while Transformers approach numeric tasks by recomputing quantities contextually at each step. Additionally, these symbol-like representations exhibit a graded nature and are influenced by task structure and model size.

**Strengths:**

1. The motivation of the paper is reasonable, and the description is very clear, making the thought process easy to follow. It’s worth mentioning that although I am not familiar with the field of causal abstraction, reading the paper and reviewing the related work allowed me to gain a general understanding of the field and appreciate the contributions of this work, which is commendable.

2. The experimental design appears to be reasonable and thorough. It identifies different characteristics of symbolic program approximation across various model architectures and analyzes the impact of model size and intervention magnitude on IIA.

**Weaknesses:**

1. Although the content of the paper is acceptable to me, I feel a bit disappointed that the paper only briefly mentions leaving more complex tasks and larger models for future research. I am unsure whether the current research approach has enough potential for further expansion.

2. As stated in Section 4.3, the reason why the Same-Object task causes poorer alignment in recurrent neural networks compared to both the Single-Object and Multi-Object tasks is unclear.

**Questions:**

Regarding the non-zero dimensions in D: How significantly does this hyperparameter impact the results?

(PS:I am not familiar with this field, though I have a relatively strong understanding of the paper’s content and personally appreciate it—I believe the clarity of this paper’s description allows even those unfamiliar with the topic to gain valuable insights. However, since I may not have read some of the most relevant related research and cannot assess whether the related work is sufficiently comprehensive, I am not in a position to accurately gauge the contribution level of this work. I trust that the Area Chair will make an appropriate decision.)

---

### Official Review · Reviewer_HvPX · 2024-11-04

**Soundness:** 2
**Presentation:** 2
**Contribution:** 2
**Rating:** 5
**Confidence:** 4

**Summary:**

This paper investigates whether neural networks, specifically GRUs, LSTMs, and Transformers, can develop symbol-like number representations through next-token prediction (NTP) tasks. Using numeric equivalence tasks as a foundation, the authors apply causal analysis, notably Distributed Alignment Search (DAS), to explore whether the models’ hidden representations align with symbolic counting behaviors, expressed by 3 hypothetical programs: Up-Down, Up-Up, and Ctx-Distr. Experimental results conclude that RNNs mimic symbolic counting by incrementing or decrementing a variable; Transformers compute context step-by-step without cumulative counter states; and larger models produce less graded alignment than larger ones.

**Strengths:**

- Novel application of DAS to study the symbolic counting behaviors of RNNs and Transformers
- The experiments present interesting results to verify the hypothesis

**Weaknesses:**

- The idea that symbol-like variables can emerge purely from next-token prediction objectives in neural networks is not surprising. Given the success of large language models (LLMs) trained with next-token prediction (NTP), numerous recent studies have empirically and theoretically validated NTP's effectiveness as a universal learning approach for many tasks including numerical reasoning [1,2].
- The paper merely highlights the alignment between the hypothesis program and the neural network representations, which does not strongly guarantee that this is the actual program the models follow. Recent research on symbolic emergence in RNNs, including studies on grammatical structures and counting without explicit symbol training, demonstrates that these neural architectures often simulate symbolic programs [3, 4]. Compared to this paper, previous works are more theoretically rigorous.
-  Similarly, the paper's findings on the counting capabilities of Transformers are superficial. Recent studies have provided a more comprehensive and systematic analysis of this ability [5], which diminishes the paper's overall contribution.
- The paper focuses solely on a basic counting task, which weakens its broader claim of discovering "symbol-like variables." To substantiate this claim, more complex symbolic tasks should be explored [6].

[1] Bubeck, Sébastien, Varun Chandrasekaran, Ronen Eldan, Johannes Gehrke, Eric Horvitz, Ece Kamar, Peter Lee et al. "Sparks of artificial general intelligence: Early experiments with gpt-4." arXiv preprint arXiv:2303.12712 (2023).

[2] Malach, Eran. "Auto-regressive next-token predictors are universal learners." arXiv preprint arXiv:2309.06979 (2023).

[3] Weiss, Gail, Yoav Goldberg, and Eran Yahav. "On the practical computational power of finite precision RNNs for language recognition." arXiv preprint arXiv:1805.04908 (2018).

[4] El-Naggar, Nadine, Andrew Ryzhikov, Laure Daviaud, Pranava Madhyastha, and Tillman Weyde. "Formal and empirical studies of counting behaviour in ReLU RNNs." In International Conference on Grammatical Inference, pp. 199-222. PMLR, 2023.

[5] Behrens, Freya, Luca Biggio, and Lenka Zdeborová. "Understanding counting in small transformers: The interplay between attention and feed-forward layers." arXiv preprint arXiv:2407.11542 (2024).

[6] Le, Hung, Dung Nguyen, Kien Do, Svetha Venkatesh, and Truyen Tran. "Plug, Play, and Generalize: Length Extrapolation with Pointer-Augmented Neural Memory." Transactions on Machine Learning Research, 2024.

**Questions:**

- In addition to alignment accuracy,  are there other methods to verify that the programs learned by the models are identical to those described in the paper?
- Numerous alternative programs could assist the model in solving counting tasks (as discussed in [3, 4]),why necessarily Up-down or Ctx-Distr?
- If theoretical support is lacking, additional experimental evidence through advanced alignment analyses or larger model evaluations would be highly valuable [7].

[7] Wu, Zhengxuan, Atticus Geiger, Thomas Icard, Christopher Potts, and Noah Goodman. "Interpretability at scale: Identifying causal mechanisms in alpaca." Advances in Neural Information Processing Systems 36 (2024).

---

> ### Author Response · Authors · 2024-11-24
> **Addressing referenced works and highlighting mech interp contributions**
>
> Yes, we agree that there have been many studies demonstrating NTP as an effective training objective in many settings. Our intention is to understand how the internal representations of number concepts that are never explicitly labeled in the task can perform these seemingly symbolic programs. In the work you referenced, the internal processing of the model is left largely opaque. We are left wondering about the interchangeability/mutability of high level concepts like numbers.
>
> We do mention the fact that causal methods like DAS only allow exploration of the algorithms that are causally distinct from one another. Taken from line 249 in the methods section,
>
> "It is important to note that there are an infinite number of causally equivalent implementations of
> these programs. For example, the Up-Down program could immediately add and subtract 1 from
> the Count at every step of the task in addition to carrying out the rest of the program as previously
> described. We do not discriminate between programs that are causally indistinct from one another in
> this work."
>
> Our alternative algorithm (the count-up, count-up algorithm) is used as a baseline to compare against a causally distinct algorithm. It is true that we could continually narrow the specifics of the algorithms we consider. This approach taken to the limit would leave the exact neural network. We are unsure what potential alternative, causally distinct algorithms you are referring to in your comment. The phrase "alignment with an algorithm" perhaps highlights a philosophical difference in the types of solutions we provide/propose vs the ones you are referring to in your comments and questions?  Our work is chiefly concerned with causal evidence for making high level claims about what algorithms a network is implementing.
>
> Thank you for the works you referenced on RNNs, these are great. We were not aware of either of the Weiss et al. 2018 or the El-Naggar et al. 2023 and both appear to be highly relevant. We have added them to our introduction at line 097. Our work can be viewed as a valuable extension of both their works for a number of reasons. For the Weiss et al.  2018, their focus is mainly on LSTMs. Their treatment of GRUs is used largely to make a point about the limitations of GRUs in terms of possible solutions. They do not appear to answer the question of how GRUs solve the task, nor do they seem to provide a causal analysis of the LSTMs they consider. Similarly for El-Naggar et al. 2023, the ReLU RNNs they train operate in a completely linear region and have limited state sizes which makes the authors' intended solution much more likely upon successful training.
>
> The work you reference on counting in transformers is, again, a great work that complements our own that we were not previously aware of. Behrens et al. 2024 explore counting in single layer transformers without positional encodings. The work narrows its focus to NoPE architectures which limits the possible solutions that the transformer can develop. We include a theoretical treatment of NoPE transformers in our supplement section A.4 to prove this point.  In our work, we supplement their findings by including explorations of two layer transformers with rotational positional encodings. This architectural difference allows for the possibility of solutions that use a cumulative state, and it allows for solutions that use positional information. Furthermore, the task used in Behrens et al. provides explicit labels for the numerosities that they investigate. Our task does not provide these labels which allows us to make claims about emergent number concepts.
>
> We agree that greater complexity is generally good for understanding how much one's findings might generalize to real world situations. We believe, however, that the claims in our paper are addressed despite the simplicity of the tasks we consider.

---

### Official Review · Reviewer_sMWR · 2024-11-04

**Soundness:** 3
**Presentation:** 3
**Contribution:** 2
**Rating:** 3
**Confidence:** 4

**Summary:**

This paper explores how transformer models implement a counting algorithm to predict sequences in which a token should be repeated an equal number of times to its occurrences in the prompt. They propose several candidate algorithms: counting up from zero to the target quantity, counting down from the target quantity to zero, and summing 1 or -1 for each token in the context. They analyze the model using Distributed Alignment Search and find that the most aligned algorithm is counting down from the target quantity to zero.

**Strengths:**

1. The paper furthers the body of work on interpreting how transformers complete mathematical or symbolic tasks

2. The paper is well-written, and the motivation is clearly conveyed

**Weaknesses:**

Novelty concerns: it does not seem like this paper advances the Pareto frontier of interpretability in transformer models. The task studied is very simple and involves repeating a token the same number of times as it occurred in the prompt, and though the introduction makes an attempt to distinguish it as a special study of symbolism that has not been previously explored, similar analyses exist in a wide range of the literature, on much more complicated tasks than this one.
1. The literature contains extensive study of transformers trained on mathematical tasks, such as modular addition (https://arxiv.org/abs/2301.05217, https://arxiv.org/abs/2306.17844), addition (https://arxiv.org/pdf/2310.13121), induction (https://arxiv.org/abs/2209.11895), relational reasoning (https://arxiv.org/pdf/2310.09753), etc. The cited Feng and Steinhardt paper (https://arxiv.org/pdf/2310.17191) is also a great example. These papers consider tasks more complex than the one considered in this paper, most of which also implicitly involve storing important properties of the prompt and representing numbers in a symbolic way, and in my view it's hard to see how the simple task considered in this paper somehow has properties that reveal a unique capability in transformer reasoning that these papers do not.
2. These papers also typically contain various methodological insights on how to perform interpretability, or analysis of training dynamics and "how" the models learned good algorithms, and which types of algorithms they are likely to learn, while this paper simply applies DAS, a technique from prior work, and does not contribute any new methods. This paper would benefit from a study of whether their "up-up" or "up-down" is more likely to be learned by a neural network for some systematic reason involving inductive biases, not just for the single network being studied.

**Questions:**

Do you think there is a reason networks should generally tend to learn the Up-Down algorithm rather than the others?

---

> ### Author Response · Authors · 2024-11-24
> **Addressing referenced works and novelty concerns**
>
> Thank you for each of the referenced works. An important point of our work is the goal of unifying different ways of understanding the numeric representations in the networks. As such, the referenced works' contributions enhance our work as we are presenting a way to unify each of their contributions into a single form of understanding. Furthermore, our work expands on each of the referenced works with new analyses and architectural variants for the numeric tasks. Namely, we provide a deeper analysis into GRUs, causal analyses for the GRUs, we include analyses of RoPE transformers, and we explore multiple variations of the numeric equivalence task.
>
> We agree that the tasks used in our work is not particularly complex compared to tasks used in other works. We raise the question, however, of why greater complexity can be useful for scientific discovery. In our case, many of the issues we wish to focus on are addressed through the tasks we present despite their simplicity. A large focus of our work is that simple variations in the task lead to different neural solutions.
>
> We seek to present the work as a contribution towards better understanding the nature of existing mechanistic interpretability methods and how to coordinate multiple methods for interpretability. We do move beyond DAS in the transformer models by applying attention manipulations and full hidden state vector substitutions (see revised Figure 3). We move beyond DAS in the recurrent models by providing an analysis of direct activation substitutions (see revised Figure 2). We also include in the supplement a theoretical treatment of one layer NoPE transformers. In this supplemental section, we show that the "strength value" within the attention mechanism can be used as a unit in a counting operation. We do, also, include various visualizations such as PCA and attention weights to supplement our causal interventions (see revised Figure 3).

---

### Author Response · Authors · 2024-11-24
**Rewritten to highlight novel contributions. Added multiple causal and theoretical analyses**

We thank the reviewers for their time and their thoughtful feedback. We have made extensive changes to the text in order to highlight the novelty of the work. Namely, we have diminished our focus on proving the existence of symbol-like variables, and instead, we highlight the the solution differences between attention and recurrence, the solution differences between variants of the task, the learning dynamics of the neural variables, the graded nature of the neural variables, and the importance of causal methods for understanding a neural network.

We've added a number of analyses to the main body of the text. Namely, we introduce variable length versions of the tasks, and examine resulting solution differences in RoPE transformers (see Figure 3). We have added a causal analysis on the attention outputs to localize the transformer computations (see Figure 3). We have added a direct neuron substitution experiment to highlight the importance of causal methods and the importance of the rotation matrix alignment used in DAS (see Figure 2 and the last paragraph in Section 4.1). We have also added a number of theoretical neural solutions to the counting task to help with thinking about the bridge between neural solutions and symbolic programs (see Figure 3).

---

### Meta-Review · Area_Chair_Gnv8 · 2024-12-19

**Metareview:**

The paper investigates the emergence of symbol-like numeric representations in neural networks trained on next-token prediction (NTP) tasks. The authors use causal interpretability methods and visualization techinques to analyze how different neural architectures, specifically GRUs, LSTMs, and Transformers, develop these representations when trained on numeric tasks. The key findings include that such models can develop analogs of interchangeable, mutable, latent number variables purely from NTP objectives; that transformers solve the problem in a notably different way than their recurrent counterparts; and that these variables change over the course of training.

The main strengths of the paper are: (i) adding to (and corroborating the findings of) existing literature on how language models encode and solve mathematical and symbolic tasks; (ii) a thorough and clear presentation, and a strong motivation; (iii) thorough experimental results presented in an engaging and compelling manner.

The main weaknesses are: (i) very narrow scope, limited to very specific counting tasks, which weakens its broader claim about emergence of symbol-like variables in ANNs (ii) the task studied is relatively simple, and similar analyses exist in the literature on more complex tasks, as pointed out by various reviewers.

Overall, my assessment of this paper is well summarized by reviewer J2Vq (Weakness 6): it needs more ambition. The findings, while interesting, are marginally different from similar findings in related work. Crucially, many such related works tackle a wider range of symbolic tasks (e.g., references [1-5] in Reviewers HvPX's Weaknesses). At a venue like ICLR, the bar for contribution would necessitate a similar (i.e., broader, more challenging) set of numeric tasks as case studies in this paper.

**Additional Comments On Reviewer Discussion:**

The four reviewers provided remarkably similar feedback, raising similar concerns, some of which were addressed by the authors', leaving the core concerns about novelty/generality of results mostly unresolved. In particular,

* Reviewer sMWR raised concerns about the novelty and complexity of the task, suggesting that the paper does not advance the interpretability frontier. The authors responded by highlighting the unifying nature of their work and adding new analyses, but the reviewer maintained their score.
* Reviewer HvPX noted that the emergence of symbol-like variables from NTP objectives is not surprising and suggested exploring more complex tasks. The authors added theoretical analyses and addressed referenced works, but the reviewer did not change their score.
* Reviewer bPA8 appreciated the clarity and thoroughness of the paper but was disappointed by the lack of exploration of more complex tasks. The reviewer maintained a their original (marginally above acceptance threshold) score.
* Reviewer J2Vq found the work easy to follow but lacking ambition. The reviewer suggested more aggressive causal interventions and exploring OOD generalization. The authors addressed some concerns but did not fully satisfy the reviewer, who raised their score to marginally below acceptance.

---

### Decision · Program_Chairs · 2025-01-22

Reject